# Methods to predict the timing and status of biological maturation in male adolescent soccer players: A narrative systematic review

Joseph Sullivan[1,2]*, Simon J. Roberts[1,3]*, John Mckeown[4], Martin Littlewood[1,3], Christopher McLaren-Towlson[5], Matthew Andrew[6], Kevin Enright[1,3]

**1** Research Institute for Sport and Exercise Sciences, School of Sport and Exercises, Liverpool John Moores University, Liverpool, United Kingdom, **2** Trauma and Orthopedics Department, Broadgreen Hospital, Thomas Drive, Liverpool, United Kingdom, **3** The Football Exchange, School of Sport and Exercises, Liverpool John Moores University, Liverpool, United Kingdom, **4** Everton Football Club, Finch Farm Training Complex, Finch Lane, Halewood, Liverpool, United Kingdom, **5** University of Hull, School of Sport Exercise and Rehabilitation Sciences, Cottingham Road, Hull, United Kingdom, **6** Department of Exercise and Sport Science, Manchester Metropolitan University, John Dalton Building, Manchester, United Kingdom

* J.F.Sullivan@2022.ljmu.ac.uk (JS); S.Roberts2@ljmu.ac.uk (SJR)

**Data Availability Statement:** All relevant data are within the paper and its Supporting Information files.

## Abstract

The aim of this review was to summarise the methods used to predict and assess maturity status and timing in adolescent, male, academy soccer players. A systematic search was conducted on PubMed, Scopus, Web of Science, CINAHL, Medline and SPORTDiscus. Only experimental studies including male, academy players aged U9-U18 years registered with a professional soccer club were included. The methodological quality of the included studies was assessed using guidelines from the Framework of Potential Biases. Fifteen studies fulfilled our inclusion criteria. Studies were mainly conducted in European countries (*n* = 12). In total, 4,707 players were recruited across all 15 studies, with an age range of 8–18 years. Five studies were longitudinal, two studies were mixed-method designs and eight studies were cross-sectional. Due to high heterogeneity within the studies, a meta-analysis was not performed. Our findings provided no equivalent estimations of adult height, skeletal age, or age at PHV. Discrepancies were evident between actual and predicted adult height and age at PHV. The Bayley-Pinneau [1952], Tanner-Whitehouse 2 [1983] and Khamis-Roche [1994] methods produced estimates of adult height within 1cm of actual adult height. For age at PHV, both Moore [2015] equations produced the closest estimates to actual age at PHV, and the Fransen [2018] equation correlated highly with actual age at PHV (>90%), even when the period between chronological age and age at PHV was large. Medical imaging techniques (e.g., Magnetic Resonance Imaging, X-Ray, Dual energy X-ray Absorptiometry) demonstrated high intra/inter-rater reliability (ICC = 0.83–0.98) for skeletal maturity assessments. The poor concordance between invasive and non-invasive methods, is a warning to practitioners to not use these methods interchangeably for assessing maturational status and timing in academy soccer players. Further research with improved study designs is required to validate these results and improve our understanding of these methods when applied in this target population.

**Funding:** The authors have received no funding for this article.

**Competing interests:** The authors have declared that no competing interests exist.

# Introduction

Professional football (soccer) clubs across the globe have an academy infrastructure dedicated to the identification and development of talented young players [1]. A professional soccer academy is a performance environment, where *potentially* talented youth players train, prepare and compete to attain the soccer-specific skills (e.g., technical; physical; tactical; psychological) to progress to the first (i.e., senior) team and succeed in competition [2–4]. Most youth academies operate extensive data capture systems where individual player's information is captured (e.g., training and match load, anthropometric and injury data) on a daily basis [5,6]. For example, in England and Wales the Player Management Application (PMA) is an online system used by academy science and medicine departments to record a range of information (e.g., training volume; training intensity; fitness testing results), which is then provided to the Premier League. Departments also pay particular attention to injury (i.e., incidence; type; location; burden) and anthropometric data that can be used to estimate each player's growth and maturation status [7]. This can subsequently facilitate the optimisation and implementation of appropriate injury prevention plans that are specific to a player's stage of maturation [8]. One example of how clubs integrate maturation and training load data to develop young players is presented by McBurnie *et al.,* [8]. This case study demonstrates how clubs can regularly use training load data gathered via geographical positioning systems (GPS) as a measure of external load, in combination with regular anthropometric and injury history data to generate a 'risk profile' for each player. This is used to create a 'decision-tree' process regarding the management of each player from a training load and injury risk prevention perspective [8].

During the adolescent growth spurt, changes in lower limb length and limb mass continue until peak height velocity (PHV) growth rate is achieved (take-off), at which point a deceleration and eventual cessation in height occur [9]. Male, youth soccer players typically undergo a phase of accelerated growth (i.e., 8-10cm) between 11–15 years of age, reaching PHV ~13–14 years of age [10]. The growth spurt coupled with maturity-associated variations are among some of the injury risk factors for the developing male athlete [10].

Injury incidence in youth academy soccer players competing in U16-U19 years are reported to be as high as four injuries per 1000 hours of training and match exposure, however injury risk and incidence is known to increase around reported mean ages (i.e., 13–14 years) at PHV [10,11]. Evidence of higher injury incidence, particularly microtraumatic damage to tissue (e.g., bone; muscle; tendon) during the period of PHV and increased general injury burden (15 vs 7 days) compared to pre-PHV (when the rate of growth in stature is at its slowest) [12] is reported in European youth soccer players [13]. Similarly, in a professional, male, Italian soccer academy, the highest injury incidence across academy age groups was reported in the U13 years, followed by the U15 and U14 years, corresponding to the period of PHV, yet caution is warranted when interpreting these findings as they are derived from a single club in Europe [14]. It would appear professional soccer clubs worldwide are becoming increasingly invested in monitoring injury rates and growth patterns of their players, particularly around PHV, due to the associated increases in injury risk, incidence, and severity that predispose players during this period [15,16]. Previous research has also highlighted the importance of youth soccer players remining 'injury-free' during their academy years, due to the negative implications of possible deselection and loss of athletic identity [16]. Therefore, it is hoped that with frequent monitoring of injury and maturation patterns, particularly around PHV, this will aid the design and implementation of targeted injury prevention and training load strategies [15,8], protecting and managing earlier maturing (skeletal age is older than chronological age by at least one year), maturing (skeletal age is ± one year of chronological age) and later

maturing players (skeletal age is younger than chronological age by at least one year) through the maturation process [17].

Evidence suggests early maturing players have the highest overall injury risk [18], with growth-related injuries (e.g., apophysitis) generally occurring pre and circa-PHV, whilst muscular and knee/ankle articular injuries occur post-PHV [11]. A combination of high training volume and relatively slow adaptation of muscles, tendons, and apophyses to changes in extremity length, mass, and moments of inertia caused by PHV are possible explanations for these findings [11]. Whereas earlier maturing players have heightened injury severity pre-PHV, later maturing players often suffer more burdensome injuries during adulthood [19]. This is reportedly due to the musculoskeletal and neuromuscular alterations induced by the individual variation in the timing of PHV amongst players within the same chronological age group [19]. To optimise injury epidemiology associated with growth and maturation within earlier, average, and later maturing players, performance staff employ methods to measure maturity status (the stage of maturation at the time of observation, i.e., pre-, circa-, or post-PHV) and timing (the age at which PHV occurs i.e., early, average, or late) [11].

The 'gold standard' indicator for assessing biological maturation includes assessments of skeletal age [20]. However, this method is invasive and involves radiation exposure due to medical scanning to assess the skeletal maturity of the hand/wrist (e.g., X-Ray, Dual energy X-ray Absorptiometry (DXA); Magnetic Resonance Imaging (MRI) [21,22] and requires clinical expertise when applied in youth environments. Furthermore, earlier work has demonstrated poor concordance for predictions of skeletal age using the Tanner-Whitehouse 2 and 3 [23,24] methods for the same wrist/hand scan in academy soccer players aged 11–17 years [25] whilst the Fels [26] method can reduce the estimation of skeletal age [27]. The systematic lowering of skeletal age associated with the Tanner-Whitehouse 3 [2001] vs. 2 method [1983] is reportedly as high as 1.06 years [25]. These variations can be attributed to the variance in reference samples from which the different methods were derived [28]. For example, the Greulich-Pyle [29] and Fels [1977] methods were developed in pediatric populations from high socioeconomic areas in the United States (US), while the Tanner-Whitehouse 2 [1983] method was developed using children in the United Kingdom (UK) [25]. The Tanner-Whitehouse 3 [2001] method, an extended version of the Tanner-Whitehouse 2 [1983] method, included children from the UK as well as adolescents from other well established soccer nations such as Japan, Belgium, Argentina, and Italy [25,30]. A further consideration is that these methods differ in the types of bones used for analysis within the hand/wrist. For instance, the Fels [1977] method uses the radius, ulna, short bones, and carpals to predict skeletal age whereas the Tanner-Whitehouse 3 [2001] method uses the radius, ulna, metacarpals and phalanges to provide a skeletal age assessment [28]. Further differences are observed for the statistical weighting and set of criteria for maturity indicators of bones within the hand/wrist between the Tanner-Whitehouse 2 [1983] and 3 [2001] methods to calculate skeletal age [25]. Given the apparent discrepancies that are evident with these invasive skeletal age assessments, non-invasive methods have been proposed as suitable alternatives for assessing maturational status and timing of PHV [9].

Two non-invasive methods for estimating maturity status and timing that are typically utilised in soccer academies are the percentage of estimated adult height and maturity offset methods [7]. The percentage of predicted adult height method provides an estimation of adult height and an estimate of the current height of a player relative to their predicted adult height [31]. The maturity offset method provides an estimate of time (years) away from PHV and subsequently an estimate of age at PHV [32]. For predicting age at PHV, other alternative equations are available for practitioners working with youth academy players. One equation proposed recently by Fransen *et al.* [33] has attempted to improve the precision of estimates for age at PHV by using a maturity ratio (chronological age / age at PHV) rather than a

maturity offset (chronological age–age at PHV), which is considered a more appropriate representation of the non-linear relationship between anthropometric variables and maturity offset [34]. Likewise, the Moore *et al.* [34] equations provide practitioners with other methods for predicting age at PHV and is considered a modification of the original Mirwald [2002] equation, however, the original regression equation used by Mirwald [2002] has been adjusted to create the Moore [2015] equations. Similarly, for predicting adult height, the Bayley-Pinneau [1952] method is widely used, as it aims to predict adult height from skeletal age and is based on the high correlation between skeletal ages attained from hand/wrist scans and the proportion of adult stature attained by adolescents at the time of the scan [35].

According to Towlson *et al.*, [10], the Mirwald [2002] and Khamis-Roche [1994] methods are the most used for predicting age at PHV and adult height respectively, since they are facilitated by organising bodies (e.g., the Premier League) and can be integrated into online PMAs. However, some criticisms of these methods are that they require more than two years of longitudinal growth data (e.g., total body height, annual growth velocity changes) and existing studies typically do not to track growth rate data for this amount of time [36]. Further limitations of these equation-based methods (e.g., maturity offset) is the tendency to overestimate the timing of PHV in earlier maturing players and underestimate the timing of PHV in later maturing players, although the accuracy of these methods improves if applied promptly with data inputted at regular intervals [36]. Parr *et al.*, [9] has also reported that the Khamis-Roche [1994] method has a greater prediction power compared to the Mirwald [2002] equation for predicting the timing of PHV, despite being primarily used to predict adult height. A limitation of the Khamis-Roche [1994] method is that it requires variables such as decimal age, standing height (cm), body mass (kg), and an accurate stature (cm) of both biological parents to provide an estimate of adult height. However, if parental height is unavailable, national averages of stature for men and women are used in the equation from qualified anthropometric assessments, which can potentially inflate the standard error [10].

Reliability concerns with these equation-based methods are associated with inconsistent research designs, study quality, and recruited populations [36]. Consequently, there is poor agreement between invasive and non-invasive prediction methods of maturity status and timing [20]. Nonetheless, these equation-based predictors remain the most practical option for practitioners working within professional soccer academies [7]. Despite a wealth of individual empirical studies, there is currently limited review studies that synthesise the existing literature and establish the reliability of both invasive and non-invasive methods for assessing maturational status and timing in youth, academy soccer players.

To our knowledge, only one previous systematic review exists that examines the accuracy and reliability of existing methods for predicting PHV in adolescents [36]. This review reported that radiograph-based methods appear to have the most value in predicting actual PHV and that the age of PHV can be accurately predicted in males as young as 11 years. The review by Mills *et al.*, [36] was conducted in healthy male and female adolescents from the general population and therefore, it is unknown how well these methods perform in youth, academy soccer players. Further findings from this review [36] suggest that equation-based methods offer some promise as surrogate measures of maturity status, though the reliability of these methods is unknown, and the current state of the literature makes such an investigation into the reliability of this particular method challenging, given the high levels of heterogeneity within the datasets. Therefore, the aim of the present review is to narratively summarise the reliability of method(s), both invasive and non-invasive, for assessing maturity status and timing in adolescent, male, academy soccer players.

## Materials and methods

The systematic review was conducted in accordance with the Preferred Reporting of Items for Systematic Reviews and Meta-Analyses (PRISMA) guidelines [37]. After several scoping searches, a comprehensive bibliographic search was conducted between June-September 2022 and re-run in May 2023 on the following academic databases: PubMed, Scopus, Web of Science, Cumulative Index to Nursing and Allied Health Literature (CINAHL), Medline, and SPORTDiscus. Search filters were limited to published literature in the English language, and articles that had full-text access. No filters regarding publication year were included and 'grey literature' (e.g., student dissertations or theses, and conference proceedings) were excluded from the search criteria.

### Search syntax

The specific keywords and syntax terms for each database were agreed upon between members of the research team and a university librarian; a database specialist employed to support the review process. The following syntax were entered into each of the above databases: Method* OR procedure* AND Estimat* OR predict* OR calculat* OR measur* AND "Peak height velocity" OR "PHV" OR matur* OR "biological maturation" OR "growth spurt*" OR "maturity offset*" OR "skeletal age" OR "skeletal maturity" AND Youth* OR adolescent* OR teenage* AND Football OR soccer AND player*.

### Inclusion/Exclusion criteria (PICOSS)

The review was planned around the Population, Intervention, Comparator, Outcome variables, Study design, Setting, (PICOSS) approach to capture appropriate quantitative studies. All members of the research team participated in devising the inclusion criteria (**Table 1**) and exclusion criteria (**Table 2**) for candidate studies. The aim of the inclusion criteria was to capture as many relevant studies as possible, that utilised either invasive (i.e., medical imaging or hand scans) or non-invasive (i.e., predictive equations) methods to assess maturational status and timing of male, academy soccer players from professional soccer clubs. The age range included U9-U18 players, in order to capture players residing in different stages of maturation from across the academy system, as well as within individual chronological age groups, with some players of the same chronological age group known to differ in biological age by as much as 5–6 years [8].

### Screening process

Studies meeting the inclusion criteria were imported into a bibliographic management software system (i.e., EndNote) for stage one and stage two screening. Stage one screening consisted of title and abstract reviews which were completed by the lead author following the

**Table 1. PICOSS study inclusion criteria.**

| | |
|---|---|
| **Population** | Male, academy soccer players aged U9-U18 years. |
| **Intervention/ Comparator** | Invasive/non-invasive methods used to predict or assess maturation status and timing. |
| **Outcome variables** | Maturity offset (years), age at PHV (years), skeletal age (years), maturity status, percentage of predicted adult height (%). |
| **Study design** | Longitudinal/cross-sectional, prospective/retrospective randomised control trials, cohort studies, case studies. |
| **Study settings** | Professional soccer club academies worldwide. |

**Table 2. Study exclusion criteria.**

| | |
|---|---|
| **Population** | Female players. Amateur/non academy players. Adolescents from the general population. |
| **Age** | Academy players aged <U9 years or > U18 years. |
| **Study characteristics** | Non-English language published studies. Descriptive/anecdotal studies. Studies based on 'expert' opinion. Non-peer reviewed articles. |
| **Outcome variables** | Soccer-specific performance characteristics (i.e. passing, shooting, tackling). Physical performance characteristics (i.e. $VO_2$ max, high-speed running distance, physical strength measures). |

removal of duplicates. Two reviewers independently screened a random sample of 20 studies and inconsistencies were resolved by consensus. Stage two screening was conducted by the first author, whereby full-text papers were assessed against the eligibility criteria. Reasons for study exclusion included those that were irrelevant to study question, inappropriate study populations, outcome variables and study designs (**Fig 1**).

## Risk of bias assessment

According to Mlinaric *et al.*, [38] the threat of publication bias in academic research is increasing, with a preference of current medical and scientific literature to publish seemingly more positive study results. Furthermore, this bias could be because more 'successful' and 'productive' studies are more interesting to read and are therefore perceived as being more valuable for publishers, editors, and their audience. To offset this threat in the present review, a thorough and objective-based inclusion criteria was provided, which used a variety of databases to capture as many relevant studies as possible, all data was considered for analysis within each study and any missing data was requested by the researchers. All included studies were quality assessed against a recognised objective framework (A Measurement tool for Assessment of Multiple Systematic Reviews) [39] to assess for risk of bias. Studies were independently assessed for risk of bias by two members of the research team with any disagreements being resolved via a discussion and no arbitrary third assessor was required.

## Quality appraisal

The quality of each study was assessed using the Framework of Potential Biases [40], which has six criteria to assess for study bias, followed by a total quality score. Quality criteria is based on: (1) study population; (2) study attrition; (3); use of valid and reliable instruments for predictors; (4) having objectively measured outcome variables; (5) controlling for confounding factors (age, gender etc.); and (6) using appropriate statistical analyses. If a criterion is fully satisfied, it receives a score of two, if the criterion is partly satisfied a score of one is given, and if the criterion is not satisfied it receives a score of zero. The score for each individual criterion is then added up to provide a total quality score for each study. A *low*-quality study has a score ranging from 0–4 points, a *medium*-quality study has a score ranging from 5–8 points and a *high*-quality study has a score ranging from 9–12 points.

## Data extraction

Extracted data for individual study outcome variables were included but were not limited to: Pearson and Spearman-rank correlational values ($R^2$), kappa and intra or inter-class coefficient values, mean differences between observed and predicted maturational status and timing

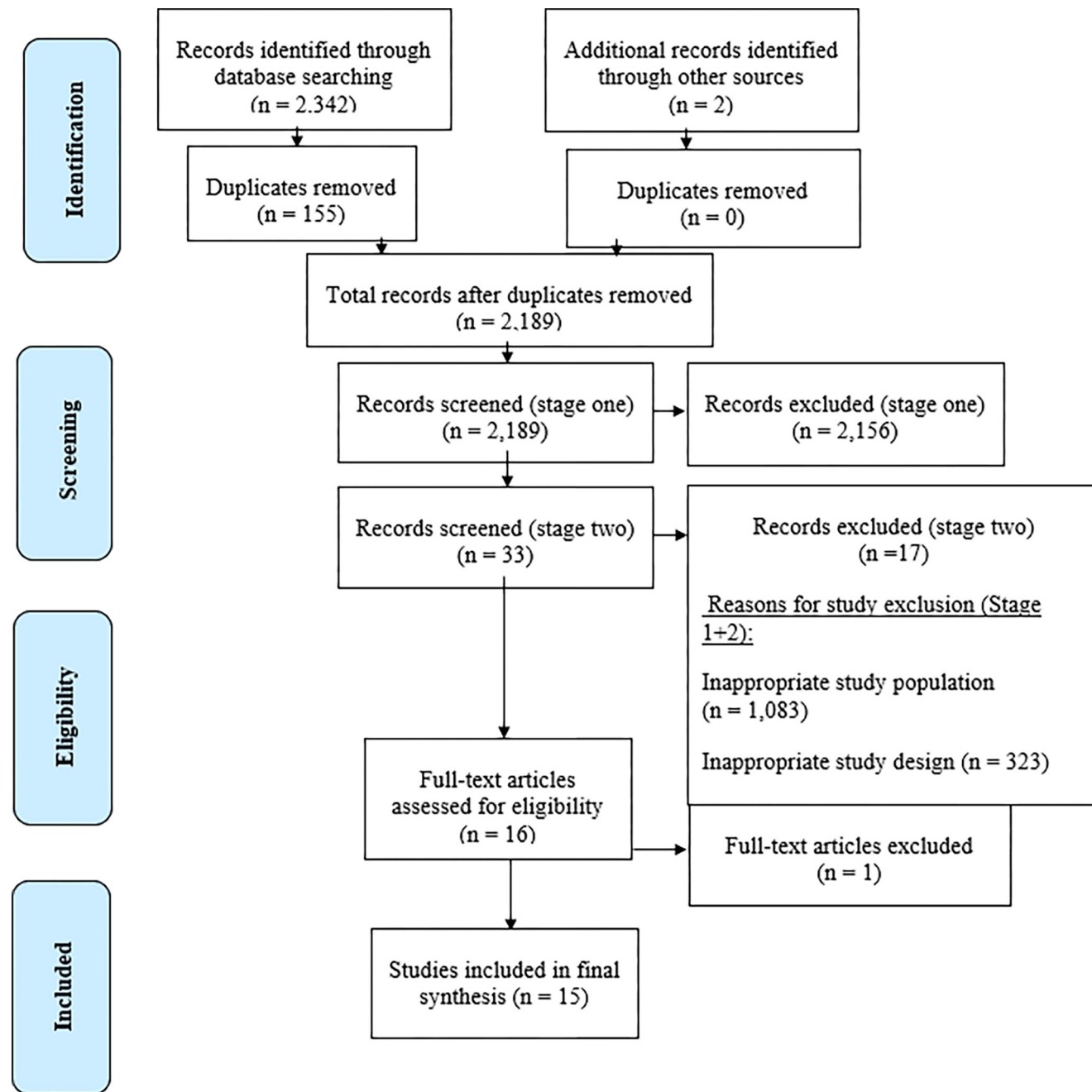

**Fig 1. PRISMA flow diagram (Moher et al., [37]).**

variables (years/cm), level of concordance between invasive and non-invasive maturity estimates and between methods for maturity classification (%).

## Data analysis

Due to assumptions of homogeneity not being satisfied and a high amount of heterogeneity within the data, a full meta-analysis was not performed [41]. A high level of heterogeneity within the data was caused by variance within individual study characteristics (cross sectional

vs. longitudinal designs), types of study data (dichotomous vs. continuous), and differences in outcome measures from individual studies (invasive vs. non-invasive outcome variables). Moreover, the vast differences in the number of included participants between studies, individual characteristics of these players not being available, and lack of reported randomisation process for included study participants also made a meta-analysis inappropriate [40]. Due to these issues, a narrative review was preferred for the study.

## A measurement tool for assessment of multiple systematic reviews (AMSTAR) 2

Previous work by Shea *et al*., [39] have commented that systematic reviews are subject to a range of biases due to the inclusion of non-randomised intervention studies, similar to the present review. A measurement tool for assessment of multiple systematic reviews (AMSTAR) model was developed to evaluate systematic reviews that utilised randomised studies [42] however it has since been updated (AMSTAR 2) to evaluate systematic reviews that have utilised non-randomised studies. The revised AMSTAR 2 tool has 16 items in total, consisting of binary '*yes*' or '*no*' questions relating to the quality of the systematic review, however it is not intended to generate an overall score.

## Narrative synthesis of findings

In total, 15 publications fulfilled our inclusion criteria and were included in the final analysis. A group summary regarding participant recruitment, study design, outcome variables and country of origin of the included studies can be found in **Table 3**.

### Quality scores

An appraisal of study quality using the framework proposed by Rashid *et al*., [40], revealed that nine studies were medium quality (5–8 points), and six studies were high quality (9–12 points), with no studies deemed low quality (0–4 points). See **Table 4**.

### AMSTAR 2

In accordance with the framework and suggestions by Shea *et al*., [39], the current review can be considered of moderate quality. The AMSTAR 2 assessment revealed that the current review contains more than one non-critical weakness (item 10 and 16, see **Table 5**) but no

**Table 3. Narrative group summary of included studies.**

| | |
|---|---|
| **Total number of participants** | $n$ = 4,707 players across 15 studies. |
| **Age range** | 8–18 years. |
| **Country of origin** | Non-European (i.e. Qatar, Brazil, Thailand, Japan, Mexico, Egypt) $n$ = 3.<br>European (i.e. Germany, UK, Belgium, Switzerland) $n$ = 12. |
| **Study design** | Cross-sectional ($n$ = 8).<br>Longitudinal ranging from five playing seasons up to 14 years ($n$ = 5).<br>Mixed-method ($n$ = 2).<br>Entirely invasive ($n$ = 1).<br>Entirely non-invasive ($n$ = 5).<br>Invasive AND non-invasive combination ($n$ = 9). |
| **Outcome variables** | Skeletal age ($n$ = 10).<br>Predicted adult height/ percentage of predicted adult height ($n$ = 6).<br>Maturity ratio ($n$ = 3).<br>Maturity offset ($n$ = 8). |

**Table 4. Study quality assessment of included studies, in accordance with the Framework of Potential Biases [40].**

| Study | Representative sample of relevant population | Study attrition (loss to follow-up and response rate) | Valid and reliable instruments for predictors | Objectively measured outcomes | Controlled for age | Appropriate statistical analyses | Quality score |
|---|---|---|---|---|---|---|---|
| Abdelbary et al., [43] | 1 | 0 | 2 | 1 | 2 | 1 | 7 Moderate |
| Fransen et al., [33] | 1 | 0 | 0 | 2 | 2 | 2 | 7 Moderate |
| Leyhr et al., [44] | 1 | 0 | 1 | 2 | 2 | 2 | 8 Moderate |
| Lolli et al., [45] | 2 | 0 | 2 | 2 | 2 | 1 | 9 High |
| Malina et al., [27] | 2 | 0 | 1 | 1 | 2 | 2 | 8 Moderate |
| Malina et al., [17] | 1 | 0 | 1 | 1 | 2 | 2 | 7 Moderate |
| Malina et al., [20] | 1 | 0 | 1 | 1 | 2 | 1 | 6 Moderate |
| Malina et al., [25] | 2 | 0 | 1 | 1 | 2 | 2 | 8 Moderate |
| Malina et al., [46] | 2 | 2 | 1 | 2 | 2 | 2 | 11 High |
| Parr et al., [9] | 2 | 0 | 1 | 2 | 2 | 2 | 9 High |
| Romann and Fuchslocher [47] | 2 | 2 | 1 | 1 | 2 | 1 | 9 High |
| Romann and Fuchslocher [22] | 2 | 2 | 1 | 0 | 2 | 2 | 9 High |
| Ruf et al., [48] | 2 | 0 | 1 | 2 | 2 | 2 | 9 High |
| Salter et al., [7] | 2 | 0 | 0 | 2 | 1 | 1 | 6 Moderate |
| Teunissen et al., [49] | 2 | 0 | 0 | 2 | 2 | 2 | 8 Moderate |

2 = criterion satisfied, 1 = criterion partly satisfied, 0 = criterion is not satisfied/ cannot be determined.

Maximum quality score = 12.

0–4 points = low quality, 5–8 points = medium quality, 9–12 points = high quality.

critical flaws. Therefore, the current review provides an accurate summary of the results from the included studies.

## Discussion

### Performance of invasive and non-invasive maturity indicators

Three of the included studies investigated adult height, with two of these studies [10,49] comparing predicted adult height methods (e.g., Khamis-Roche, 1994; Bayley-Pinneau, 1952) to actual (observed) adult height and one study [48] compared two predictive methods for adult height (**Table 6**). The findings revealed that none of the methods produced the same estimation of adult height, with discrepancies (-0.45cm to -2.1cm) evident between predicted and observed values of adult height. One plausible explanation for these discrepancies could be attributed to the methods employed during the anthropometric data collection within the studies. For example, it is unclear whether anthropometric data was captured under

**Table 5. AMSTAR 2 systematic review assessment.** Responses in bold are considered the key domains as suggested by Shea *et al.*, [39].

| Criterion | Response |
|---|---|
| *1. Did the research question and inclusion criteria for the review include the components of PICO?* | Yes |
| **2.** *Did the report of the review contain an explicit statement that the review methods were established prior to the conduct of the review and did the report justify any significant deviations from the protocol?* | **Yes** |
| *3. Did the review authors explain their selection of the study designs for inclusion in the review?* | Yes |
| *4. Did the review authors use a comprehensive literature search strategy?* | **Yes** |
| *5. Did the review authors perform study selection in duplicate?* | Yes |
| *6. Did the review authors perform data extraction in duplicate?* | Yes |
| *7. Did the review authors provide a list of excluded studies and justify the exclusions?* | **Yes** |
| *8. Did the review authors describe the included studies in adequate detail?* | Yes |
| *9. Did the review authors use a satisfactory technique for assessing the risk of bias (RoB) in individual studies that were included in the review?* | **Yes** |
| *10. Did the review authors report on the sources of funding for the studies included in the review?* | No |
| *11. If meta-analysis was performed, did the review authors use appropriate methods for statistical combination of results?* | **N/A** |
| *12. If meta-analysis was performed, did the review authors assess the potential impact of RoB in individual studies on the results of the meta-analysis or other evidence synthesis?* | N/A |
| *13. Did the review authors account for RoB in primary studies when interpreting/discussing the results of the review?* | **Yes** |
| *14. Did the review authors provide a satisfactory explanation for, and discussion of, any heterogeneity observed in the results of the review?* | Yes |
| *15. If they performed quantitative synthesis did the review authors carry out an adequate investigation of publication bias (small study bias) and discuss its likely impact on the results of the review?* | **Yes** |
| *16. Did the review authors report any potential sources of conflict of interest, including any funding they received for conducting the review?* | No |

International Society for the Advancement of Kinanthropometry (ISAK) guidance or by single or multiple measurer(s) [9,45]. Such methodological considerations may elevate the poor level of agreement amongst these predictive methods [20]. This finding is of relevance and importance to practitioners in academy soccer (e.g., sport scientists) given the application of predicted adult height to categorise players into maturity specific groupings (i.e., 'bio-banding') [50]. Thus, erroneous predictions of player maturity status may incur the mis-categorisation of players into such groupings and afford players with unfair playing environments (e.g., competing against players who matured earlier and subsequently possess enhanced anthropometric characteristics, or vice-versa). Careful consideration should be taken by practitioners attempting to assess maturational status in academy soccer players.

The evidence in this review suggests the Bayley-Pinneau [1952], Tanner-Whitehouse 2 [1983], and Khamis-Roche [1994] predictive methods performed well against observed adult height and produced estimates within 1cm of actual adult height. However, Tanner-Whitehouse 3 [2001] estimates were 1–2 cm short of observed adult height. For predictive estimates, large agreements and small systematic errors were observed between the Tanner-Whitehouse 2 [1983] and Khamis-Roche [1994] methods, which demonstrates the high level of concordance between these predictive methods [48]. The Khamis-Roche [1994] method produced a slightly higher estimate of adult height compared to the Tanner-Whitehouse 2 [1983] method, with an observable difference of around 0.73cm, which is considered acceptable [48]. One potential reason for the observed differences between methods could be the different player nationalities. Two studies were conducted in Europe (England and Germany) whilst the other study was conducted in Qatar. These different nationalities and ethnicities could play a key

**Table 6. Performance of invasive (i.e. medical imaging and hand scans) and non-invasive (i.e. predictive equations) maturity indicators.**

| | | | | Adult height | | |
|---|---|---|---|---|---|---|
| Study | Population | Population demographics | Country | Observed/ predicted | Method | Data |
| Lolli *et al.*, [45] | N = 103 youth academy players | Age (years): 11–17 at time of data collection.<br>Sex: Male<br>Body mass: Not reported<br>Height (cm): 137.5–187 | Qatar | Observed vs predicted | Actual—predicted adult height (BoneXPert, TW2, TW3) | <u>Chronological age 12.5–17.5 years:</u><br>BoneXPert (Bayley-Pinneau): -0.46cm<br>TW2: -0.45cm<br>TW3: -1.32cm<br><u>Skeletal age 12.5–17.5 years:</u><br>BoneXPert (Bayley-Pinneau): -0.89cm<br>TW2: -0.53cm<br>TW3: -2.1cm |
| Parr *et al.*, [9] | N = 23 youth soccer players | Age (years):<br>Initial observation: 12.4 ± 0.6<br>Final observation: 15.4 ± 0.6<br>Sex: Male<br>Height (cm):<br>U13: 162.2 ± 7.6<br>U14: 167.8 ± 8.1<br>U15: 175.5 ± 7.0<br>U16: 178.8 ± 4.6<br>U17: 179.2 ± 4.2<br>Body mass: Not reported<br>Ethnicity: N = 15 European N = 8 non-European | England | Observed vs predicted | Predicted (Khamis-Roche)—observed adult height | -0.9 cm |
| Ruf *et al.*, [48] | N = 114 youth soccer players | Age (years):<br>U12: 11.4 ± 0.3<br>U13: 12.6 ± 0.3<br>U14: 13.5 ± 0.2<br>U15: 14.6 ± 0.3<br>U16: 15.5 ± 0.4<br>U17: 16.5 ± 0.4<br>Sex: Male<br>Height (cm):<br>U12: 146.4 ± 6.2<br>U13: 153.5 ± 6.6<br>U14: 167.0 ± 8.3<br>U15: 171.1 ± 5.8<br>U16: 177.9 ± 6.6<br>U17: 174.7 ± 6.6<br>Body mass (kg):<br>U12: 37.3 ± 6.2<br>U13: 41.3 ± 5.1<br>U14: 56.6 ± 9.4<br>U15: 61.2 ± 7.4<br>U16: 69.0 ± 7.5<br>U17: 70.3 ± 7.3<br>Ethnicity: European, African, Middle-eastern | Germany | Predicted vs predicted | BAUS (TW2)—Khamis- Roche: Predicted adult height difference Percentage of predicted adult height difference | -0.73cm<br>0.37% |
| | | | | Skeletal age | | |
| Study | Population | Population demographics | Country | Observed/ predicted | Method | Data |

(*Continued*)

**Table 6.** (Continued)

| | | | | Adult height | | |
|---|---|---|---|---|---|---|
| **Study** | **Population** | **Population demographics** | **Country** | **Observed/ predicted** | **Method** | **Data** |
| Malina et al., [25] | N = 1,831 youth soccer players | Age (years): 10–17 Body mass: Not reported Height: Not reported | Portugal Belgium Spain Japan Thailand Italy Mexico Brazil | Predicted vs predicted | TW3 –TW2 SA difference | 11 years: -0.97 years 12 years: -1.13 years 13 years: -1.16 years 14 years: -1.09 years 15 years: -1.02 years 16 years: - 1.00 years 17 years: -1.07 years Total: -1.06 years |
| | | | | Age at PHV | | |
| **Study** | **Population** | **Population demographics** | **Country** | **Observed/ predicted** | **Method** | **Data** |
| Parr et al., [9] | N = 23 youth soccer players | Age (years): Initial observation: 12.4 ± 0.6 Final observation: 15.4 ± 0.6 Sex: Male Height (cm): U13: 162.2 ± 7.6 U14: 167.8 ± 8.1 U15: 175.5 ± 7.0 U16: 178.8 ± 4.6 U17: 179.2 ± 4.2 Body mass: Not reported Ethnicity: European and Non-European | England | Observed vs predicted | Predicted (Mirwald)—observed age at PHV | 0.89 years |
| Teunissen et al., [49] | N = 17 youth soccer players | Age (years): 11.9 ± 0.8 Sex: Male Height (cm): 149.7 ± 6.2 Body mass (kg): 38.9 ± 5.9 Ethnicity: European ancestry, African, Middle-Eastern | Netherlands | Observed vs predicted | Observed–predicted (Mirwald, Moore 1, Moore 2, Fransen) age at PHV | Observed age at PHV = 13.8 years Mirwald: 0.6 years Moore 1: 0.6 years Moore 2: 0.3 years Fransen: 0.7 years |
| Fransen et al., [33] | N = 1,330 youth soccer players | Age (years): 8–17 Sex: Male Body mass: Not reported Height: Not reported Ethnicity: Diverse, mainly Caucasian | Belgium | Predicted vs predicted | Maturity ratio vs maturity offset predictions of age at PHV | Maturity offset Standard error: 1.962 Correlation: 89.22% Maturity ratio Standard error: 0.051 Correlation: 90.19% |
| | | | | Biological age | | |
| **Study** | **Population** | **Population demographics** | **Country** | **Observed/ predicted** | **Method** | **Data** |
| Salter et al., [7] | N = 113 youth soccer players | Age (years): 14.3 ± 1.1 Sex: Male Height (cm): 170.1 ± 10.6 Body mass (kg): 58.7 ± 10.5 Ethnicity: 90% White-British, <10% from other ethnic minorities | England | Predicted vs predicted | Mirwald vs Moore vs Fransen vs Khamis-Roche | Mirwald: 14.4 years Moore: 14.3 years Fransen: 14.3 years Khamis-Roche: 14.7 years |

*TW2 = Tanner-Whitehouse 2. TW3 = Tanner Whitehouse 3*.

part in growth variables, such as proportions of sitting height and leg length to stature ratio, which are known to vary among ethnic/racial groups and thus could influence the difference between observed and predicted adult height values derived from different methods [20]. For example, previous work from Lopez *et al*., [51] concluded that adolescent soccer players in Chile were smaller and lighter than the general South American population for any given age within adolescence and demonstrate lower growth rates compared to Brazilian and Spanish soccer players. Given that many of these existing predictive methods were derived using White and Caucasian populations from middle-class backgrounds, the appropriateness of using current predictive equations for adult height in culturally and ethnically diverse environments (e.g., professional soccer academies) currently remains unknown [7]. Therefore, validation of existing equations or proposal of new equations in this target population may be more appropriate for soccer practitioners to use to assess and estimate adult height in youth players.

One study investigated the use of Tanner-Whitehouse 2 [1983] and Tanner-Whitehouse 3 [2001] methods to predict skeletal age [25]. The finding of this study was that Tanner-Whitehouse 3 [2001] skeletal ages were on average 1.06 years younger than skeletal ages derived from the Tanner-Whitehouse 2 [1983] method across the U12 to U17 age groups. The difference between Tanner-Whitehouse 2 [1983] and Tanner-Whitehouse 3 [2001] skeletal ages was greatest between the U12 to U14 age groups, a significant period during maturation that is associated with rapid increases in skeletal and somatic growth [52]. Given the systematic lowering of skeletal ages associated with the Tanner-Whitehouse 3 [2001] method vs. Tanner-Whitehouse 2 [1983] method, this could elevate the risk of incorrect maturity classification of players [25], as well as having implications for bio-banding in soccer tournaments, leading to players potentially being incorrectly included or excluded in tournaments with peers of a similar skeletal or chronological age [17]. These observed differences between Tanner-Whitehouse 2 [1983] and Tanner-Whitehouse 3 [2001] methods could be explained by the reference samples used to derive the estimates of skeletal age associated with each method. For example, the Tanner-Whitehouse 2 [1983] method was developed in children in the UK, unlike the Tanner-Whitehouse 3 [2001] method, which used a more heterogenous sample of children from Spain, Italy, Belgium, Argentina, and Japan [45]. The differences between the populations used to derive these skeletal age estimates could partly explain the variance. According to Malina and Bouchard [53] skeletal maturation in Hispanic adolescents occurs later than similarly aged Black and White adolescents. Furthermore, Asian adolescents appear to be, on average, shorter, lighter and are likely to be more skeletally immature compared to similarly aged adolescents of European ancestry. Given these differences in maturational growth patterns between adolescents of various ethnicities used within the reference samples, it is unsurprising that the Tanner-Whitehouse 2 and 3 [1983; 2001] methods produce inequivalent estimates of skeletal age. Other possible explanations for these different skeletal ages could be due to the differences in the criteria for maturity indicators and the associated statistical weighting provided to maturity indicators being different between Tanner-Whitehouse 2 and 3 [1983; 2001] methods, ultimately deriving different skeletal ages [17]. Thus, the most reliable method for estimating skeletal age remains unclear, yet the current review is supportive of claims by Malina *et al*. [25] who advocated using Tanner-Whitehouse 2 [1983] rather than 3 [2001] due to the systematic lowering of skeletal ages and the potential negative consequences this may have during maturational assessments when using the latter method.

Our findings suggest none of the estimated ages at PHV were equivalent to actual ages at PHV with any of the proposed predictive methods, which does question the precision of these methods [Teunissen *et al*., 49]. On average, the Moore 2 [2015] equation estimate was the closest to actual age at PHV (mean range = 0.3 years), followed by Moore 1 (2015, mean range = 0.6 years), Fransen (2018, mean range = 0.7 years), and Mirwald (2002, mean

range = 0.75 years). As these are group and not individual estimates, caution must be taken when interpreting these findings, given that large inter-individual differences in maturational timing are evident between players of the same chronological age group [9]. Recent work from Teunissen *et al.* [49] reported the Mirwald [2002] and Fransen [2018] equations provide the most stable estimates of age at PHV over time, though none of these equations have longitudinal stability in more than 45% of players, with evidently wide 95% confidence intervals (Fransen, 2018 = -0.38–0.25 years; Mirwald, 2002 = -0.29–0.12 years). The Fransen [2018] equation demonstrated higher correlative values with actual age at PHV compared to the Mirwald [2002] equation (90% vs. 89% respectively), even when the difference between age at PHV and observed chronological age was large [32]. This could provide some confidence to practitioners aiming to predict age at PHV in academy youth soccer players, as this equation can be applied from an early chronological age without inflating the prediction error, though more research is needed to support this claim. Recent criticism of the Fransen [2018] method has emerged, which soccer practitioners attempting to use this method need to carefully consider. According to Nevill and Burton [54], the Fransen [2018] method is flawed due to the inclusion of a player's chronological age in both sides of the prediction equation, which the authors argue will inevitably result in high correlative $R^2$ values similar to the present review. Given that age at PHV is used by many professional soccer club academies to assess their players [7], it is worth noting that all predictions have associated errors when applied to individual players and therefore, the individual timing and rate of growth spurt need to be considered for all players when selecting the appropriate predictive method to use for deriving age at PHV on an individual basis [9]. From a group perspective, both Moore 1 and 2 [2015] equations produced the smallest amount of over/underestimation (0.3 and 0.6 years respectively) from the observable age at PHV.

One study investigated biological age amongst four predictive equations [7]. All four equations were consistent in their estimates for biological age with a maximum difference of 0.3 years, suggesting that there are tight limits of agreement. The tight limits of agreement between the maturity offset methods [Fransen, 2018; Mirwald, 2002; Moore, 2015] is unsurprising given they all derived from the same original regression equation. Still, the percentage of adult height equation was derived from a different regression equation [31], therefore this could be an underlying reason for the higher biological age with this method compared to the previous three [7]. Furthermore, the Khamis-Roche [1994] method contains a genetic component within the equation by including mid-parental height, a variable that is not used with the other equations, which could also explain the slight difference in biological age using the Khamis-Roche [1994] method in comparison to the maturity offset methods. One criticism of this study is the lack of inclusion for any observed values of biological age to compare these estimates against, therefore, the true reliability of these predictive methods remains unknown. One final suggestion proposed by Salter *et al.* [7], which the present review supports, is to not use maturity offset methods and predicted adult height methods interchangeably, given they provide different estimates of biological age.

## Concordance between invasive and non-invasive methods

One of the major aims of the present review was to evaluate the level of agreement between invasive and non-invasive methods for assessing maturational status and timing in academy soccer players. Previously, relatively poor agreement between invasive and non-invasive methods for assessing maturity status has been reported [20] and the present review supports this supposition. Findings suggest a moderate agreement, at best, between invasive and non-invasive methods for assessing maturational status and timing (**Table 7**). Due to the lack of

**Table 7. Concordance between invasive and non-invasive methods.**

| Study | Population | Population demographics | Country | Method | Correlation | Kappa coefficient | Percentage of agreement | Magnitude of agreement |
|---|---|---|---|---|---|---|---|---|
| Lehyr et al., [44] | N = 63 German soccer players | Age (years): U12: 11.3 ± 0.3 U14: 13.4 ± 0.3 Sex: Male Body mass (kg): U12: 39.13 ± 4.33 U14: 51.37 ± 8.88 Height (cm): U12: 150.06 ± 5.48 U14: 164.86 ± 10.23 | Germany | U12 SA MRI vs SA US SA MRI vs Mirwald SA MRI vs Khamis-Roche U14 SA MRI vs SA US SA MRI vs Mirwald SA MRI vs Khamis-Roche Total SA MRI vs SA US SA MRI vs Mirwald SA MRI vs Khamis-Roche | 0.56 0.63 0.66 0.65 0.74 0.61 0.80 0.84 0.81 | Not reported | Not reported | Not reported |
| Malina et al., [20] | N = 180 youth soccer players | Age (years): 10–15 Sex: Male Height: Not reported Body mass: Not reported | Portugal | 11–12 years Percentage of predicted adult height vs SA-CA difference Age at PHV vs SA-CA difference Age at PHV vs Percentage of predicted adult height: SA-CA vs pubic hair stages 1–5 Age at PHV vs pubic hair stages 1–5 Percentage of predicted adult height vs pubic hair stages 1–5 13–14 years Percentage of predicted adult height vs SA-CA difference Age at PHV vs SA-CA difference Age at PHV vs Percentage of predicted adult height SA-CA vs pubic hair stages 1–5 Age at PHV vs pubic hair stages 1–5 Percentage of predicted adult height vs pubic hair stages 1–5 | 0.27 0.43 0.26 0.40 0.50 0.36 0.47 0.29 0.34 0.40 0.16 0.34 | 0.23 0.11 0.12 Not reported Not reported Not reported Not reported 0.23 0.13 0.02 Not reported Not reported Not reported | 57% 55% 75% Not reported Not reported Not reported 63% 57% 61% Not reported Not reported Not reported | Not reported Not reported Not reported Not reported Not reported Not reported Not reported Not reported Not reported Not reported Not reported Not reported |
| Romann & Fuchoslacher [47] | N = 119 youth soccer players N = 6 national coaches of U15-21 Swiss national team | Age (years): 14 ± 0.3 Sex: Male Height (cm): 164.9 ± 8.4 Body mass (kg): 53 ± 8.4 | Switzerland | Skeletal age vs coaches eye Skeletal age vs age at PHV | 0.62 0.42 | 0.48 0.25 | 73.9% 65.5% | Moderate Fair |

(*Continued*)

**Table 7.** (Continued)

| Study | Population | Population demographics | Country | Method | Correlation | Kappa coefficient | Percentage of agreement | Magnitude of agreement |
|-------|-----------|------------------------|---------|--------|-------------|-------------------|-------------------------|------------------------|
| Ruf *et al.,* [48] | N = 114 youth soccer players | Age (years): U12: 11.4 ± 0.3 U13: 12.6 ± 0.3 U14: 13.5 ± 0.2 U15: 14.6 ± 0.3 U16: 15.5 ± 0.4 U17: 16.5 ± 0.4 Sex: Male Height (cm): U12: 146.4 ± 6.2 U13: 153.5 ± 6.6 U14: 167.0 ± 8.3 U15: 171.1 ± 5.8 U16: 177.9 ± 6.6 U17: 174.7 ± 6.6 Body mass (kg): U12: 37.3 ± 6.2 U13: 41.3 ± 5.1 U14: 56.6 ± 9.4 U15: 61.2 ± 7.4 U16: 69.0 ± 7.5 U17: 70.3 ± 7.3 Ethnicity: European, African, Middle-eastern | Germany | Z score 0.50: Percentage of predicted adult height vs SA-CA difference Z score 0.75: Percentage of predicted adult height vs SA-CA difference Z score 1.00: Percentage of predicted adult height vs SA-CA difference BAUS software vs Khamis-Roche: Predicted adult height Percentage of predicted adult height Biological age | 0.52 0.49 0.45 0.86 0.96 0.80 | 0.37 0.39 0.31 Not reported Not reported Not reported | 65% 68% 66% Not reported Not reported Not reported | Not reported Not reported Not reported Not reported Not reported Not reported |

*MRI = Magnetic Resonance Imaging. US = Ultrasound. SA = Skeletal age. CA = Chronological age*.

concordance between invasive and non-invasive methods, caution is required when interpreting correlative values based on non-significant and significant Spearman or Pearson factors. One criticism of the studies that investigated the agreement between invasive and non-invasive methods, is the over reporting of correlative values and inconsistent reporting of the size of agreement between these methods [20,44,48], in addition to a lack of longitudinal follow up on the true relationship between these methods [20,44,47,48]. The studies in this review were largely inconsistent in the reporting of effect sizes, therefore the true nature of the relationship (s) cannot be determined with confidence.

Invasive methods are considered as, the 'gold standard' for assessing biological maturation in adolescent soccer players [20] and therefore it was unsurprising to see that moderate to high correlations (skeletal age vs. pubic hair, $r = 0.4$; skeletal age assessed via Magnetic Resonance Imaging and ultrasound techniques, $r = 0.8$) between invasive methods existed [20]. Some differences were observable between different age groups for the concordance between invasive and non-invasive methods (e.g., U12, $r = 0.62$; U14, $r = 0.67$), with higher correlations for the concordance in the older vs. the younger age group in some studies [44] but not others (U11-U12 = 62%; U13-U14 = 60%) [20]. This could be representative of a general maturity factor associated with maximal growth and biological maturation typically seen with this older age group [20,44], moreover, it could also represent the variation in the individual timing of maturation associated with players around this period, with maturity status varying as much as 5–6 years for players of the same chronological age [8]. Despite the high correlative values, the size of agreement and associated effect sizes between invasive methods was not reported by any of the included studies and therefore further investigation is required.

The analysis of non-invasive method results revealed only fair to moderate agreement with invasive methods [20,47,48]. Similar trends occurred whereby high correlative values did not

translate into similar levels of agreement for the relationship between invasive and non-invasive methods [48]. Methods such as percentage of predicted adult height ranged from 57–68% in agreement with invasive methods (e.g., skeletal age) whereas age at PHV ranged from 55–65% in agreement with invasive methods. Two noteworthy findings from the data demonstrate a high level of agreement between two non-invasive methods (age at PHV and percentage of predicted adult height) ranging between 61–75%, possibly due to the collection of similar anthropometric variables [44], and the use of 'coaches eye' (i.e., a subjective estimation made by coaches on individual player maturity status) having moderate levels of agreement with skeletal age (74%). However, the latter finding should be viewed with some caution as this method is still prone to error and requires experienced staff members to make valid estimations of player maturation [47]. Furthermore, this study was also limited to a cross-sectional study design, so the longitudinal stability of this method is yet to be determined. The disparity between invasive and non-invasive methods may be explained by the population differences between the reference samples used for developing the non-invasive methods currently used within current professional soccer environments (e.g., Mirwald, 2002; Khamis-Roche, 1994) and modern academy youth players. The existing non-invasive equations for predicting age at PHV and adult height were mainly developed on adolescents of European ancestry from the general population [20]. However, youth, academy soccer players worldwide tend to mature earlier in comparison to adolescents from the general population after 13 years of age [55]. This advanced skeletal maturity is associated with transient increases in body mass, muscular strength and power, and $VO_2$ max [55]. Therefore, it is reasonable to conclude that academy soccer players are not equivalent to the general adolescent population, and the sample used for developing current non-invasive predictive equations [18]. These population differences question the validity and reliability of using these non-invasive methods within academy soccer players and further investigative studies that take these population differences into consideration are required. Collectively, these results indicate that invasive and non-invasive methods should not be used interchangeably given their relatively poor agreement [43], therefore practitioners are advised not to combine invasive and non-invasive methods when assessing maturational status and timing in academy soccer players [44].

## Reliability of X-Ray, DXA and MRI scanning techniques

Investigative studies regarding the reliability of invasive scanning techniques in academy soccer players remain limited, with only two studies included in the current review [22,43]. However, the findings from these studies report acceptable estimates for assessing skeletal age and maturity status in academy soccer players (**Table 8**). Inter-observer agreement was considered excellent for using DXA (intra class coefficient = 0.93) and X-Ray (intra class coefficient = 0.92) scanning to assess skeletal maturity [22]. Meanwhile, MRI (intra class coefficient = 0.828), inter-observer agreement was considered very good. On the other hand, intra-rater reliability for DXA and X-Ray were also considered excellent with intra class coefficients ranging from 0.95–0.97 for DXA and 0.98 for X-Ray, respectively. Unfortunately, no values for intra-rater reliability were reported for MRI which can be considered a limitation of the study [43]. Collectively, the results demonstrate the efficiency of MRI, X-Ray, and DXA scanning for assessing skeletal maturity in academy soccer players, yet further validation of these methods is needed in players of different ethnicities as well as longer follow-up periods to ensure long term reliability.

Despite the efficacy of these methods for assessing skeletal maturity and age in academy soccer players, subtle differences exist between the characteristics of these methods. For example, DXA scans are known to have significantly less radiation compared to MRI and X-Rays

**Table 8. Reliability of X-Ray, DXA and MRI scanning techniques.**

| Study | Population | Population demographics | Country | Outcome variables | ICC | Classification |
|---|---|---|---|---|---|---|
| Abdelbary *et al.*, [43] | N = 61 youth soccer players | Age (years): 13–18<br>Sex: Male<br>Height: Not reported<br>Body mass: Not reported | Egypt | Inter-rater reliability | MRI grade of fusion vs actual age<br>0.828 | Very good |
| Romann & Fuchoslacher [22] | N = 63 youth soccer players | Age (years): 14 ± 0.3<br>Sex: Male<br>Height (cm): 164.9 ± 8.4<br>Body mass (kg): 53 ± 8.7 | Switzerland | Inter/intra-rater reliability | Intra-rater DXA<br>R1: 0.97<br>R2: 0.95<br>Inter-rater DXA<br>R1 + R2: 0.93<br>Intra-rater X-Ray<br>R1: 0.98<br>R2: 0.98<br>Inter-rater X-Ray<br>R1 + R2: 0.92 | Excellent<br>Excellent<br>Excellent<br>Excellent<br>Excellent<br>Excellent |

*MRI = Magnetic Resonance Imaging. DXA = Dual energy X-ray Absorptiometry*.

[22] and given the similar level of agreement between X-Ray and DXA scanning for assessing skeletal maturity, practitioners in soccer may be inclined to select DXA scanning instead of X-Ray scanning. However, DXA scanning is more time-consuming and expensive compared to X-Rays, which are additional considerations for academy soccer practitioners [22].

MRI has received more research attention than X-Ray and DXA scanning for assessing skeletal maturity [21,43]. MRI correlates highly with chronological age [21] and findings from Abdelbary *et al.*, [43] support the use of MRI to assess skeletal maturity in academy soccer players. Further evidence of the use of MRI includes shorter scanning times and higher image resolution, however high costs and expertise required are potential disadvantages of this method. In sum, all the discussed invasive methods report high cross-sectional reliability for assessing skeletal maturity in academy soccer players, but further validation of these methods and exploration of other alternatives (e.g., ultrasound) are needed.

## Concordance of maturity status classification

Only a moderate agreement was found for the concordance of maturity status classifications (**Table 9**). Utilising the Fels [1977] method compared to MRI to identify skeletally mature players, revealed that more players were classed as skeletally mature using the Fels [1977] method compared to MRI across the U15-U17 years, particularly for ages 16–17 years. A combined total of players aged 16–17 years reported that 62% of players were skeletally mature utilising the Fels [1977] method, whereas only 22% were skeletally mature with MRI. Results are limited to three age groups and thus may not represent the true discrepancies between these methods within the full academy system. One explanation for these methodological discrepancies could be that MRI has six stages of fusion as criteria to describe skeletal maturity, but the Fels [1977] method only has four [17], therefore, the researcher interpretation of criteria to ascertain the degree of fusion for skeletal maturity at each stage may differ between methods, with MRI fusion described via percentages and descriptive information and the Fels [1977] method relying solely on descriptive information to assess skeletal fusion [17].

A moderate agreement (55%) between the Fels [1977] and Tanner-Whitehouse 3 [2001] method was reported in the data, with a slightly lower level of agreement reported between the Tanner-Whitehouse 2[1983] and 3 [2001] methods (52%). Differences were observed in the agreement for the number of earlier (33%) and average (86%) maturing players between the

**Table 9. Concordance of maturity status classification.**

| Study | Population | Population demographics | Country | Invasive/non-invasive | Data |
|---|---|---|---|---|---|
| Malina et al., [17] | N = 592 youth soccer players | Age (years): Series 1: 11–17 Series 2: 11–17 Series 3: 12–16 Sex: Male Height: Not reported Body mass: Not reported | Portugal, Spain | Invasive MRI grade of fusion vs Fels SA frequency of skeletally mature players | 15 years Fels: 8% MRI: 3% 16 years Fels: 23% MRI: 7% 17 years Fels: 39% MRI: 15% |
| Malina et al., [27] | N = 40 youth soccer players | Age (years): 12–16 U11-12: 12.78 ± 0.18 U13-14: 14.1 ± 0.39 U15-16: 15.7 ± 0.32 Sex: Male Height: Not reported Body mass: Not reported | Spain | Invasive Fels vs TW3 percentage of agreement | Late maturers 100% Average maturers 85.7% Early maturers 33.3% Mature 100% Correlation: 0.66 Kappa coefficient: 0.59 Percentage of agreement: 55% Magnitude of agreement: Moderate |
| Malina et al., [25] | N = 1,831 youth soccer players | Age (years): 10–17 Body mass: Not reported Height: Not reported | Portugal, Belgium, Spain, Japan, Thailand, Italy, Mexico, Brazil | Invasive TW2 vs TW3 | U11 Late: 13.4% Average: 34.5% Early: 9% Percentage of agreement: 56.9% U12 Late: 13.6% Average: 17.5% Early: 19.7% Percentage of agreement: 51.1% U13 Late: 8.7% Average: 19.3% Early: 18.4% Percentage of agreement: 46.4% U14 Late: 4.8% Average: 26.7% Early: 26.7% Percentage of agreement: 58.1% U15 Late: 4% Average: 33.3% Early: 7.3% Percentage of agreement: 44.6% Total: Late: 9.8% Average: 25.5% Early: 16.7% Percentage of agreement: 52% |
| Malina et al., [46] | N = 58 youth soccer players | Age (years): 11–14 Sex: Male Height: Not reported Body mass: Not reported Ethnicity: European ancestry | Portugal | Non-invasive Mirwald vs Moore | Early Mirwald: 0% Moore: 3% Average Mirwald: 43% Moore: 50% Late Mirwald: 66% Moore: 43% |
| Salter et al., [7] | N = 113 youth soccer players | Age (years): 14.3 ± 1.1 Sex: Male Height (cm): 170.1 ± 10.6 Body mass (kg): 58.7 ± 10.5 Ethnicity: 90% White-British, <10% from other ethnic minorities | England | Non-invasive Moore vs Fransen vs Mirwald vs Khamis-Roche | 85–96% PAH Moore–Mirwald kappa: 0.67 (substantial) Fransen–Mirwald kappa: 0.66 (substantial) Fransen–Moore kappa: 0.64 (substantial) Khamis-Roche–Mirwald kappa: 0.49 (moderate) Khamis-Roche–Moore kappa: 0.50 (moderate) Khamis-Roche–Fransen kappa: 0.44 (moderate) 88–93% PAH Moore–Mirwald kappa: 0.60 (moderate) Fransen–Mirwald kappa: 0.59 (moderate) Fransen–Moore kappa: 0.58 (moderate) Khamis-Roche–Mirwald kappa: 0.31 (fair) Khamis-Roche–Moore kappa: 0.43 (moderate) Khamis-Roche–Fransen kappa: 0.39 (fair) 85–95% PAH Maturity offset methods: 64–67% (substantial) Maturity offset vs PAH methods: 44–50% (moderate) 88–93% PAH Maturity offset methods: 58–60% (moderate) Maturity offset vs PAH methods: 31–43% (fair) |

* MRI = Magnetic Resonance Imaging. SA = Skeletal age. TW2 = Tanner-Whitehouse 2. TW3 = Tanner-Whitehouse 3. PAH = Predicted Adult Height*.

Fels [1977] and Tanner-Whitehouse 3 [2001] method and the highest level of agreement between Tanner-Whitehouse 2 [1983] and 3 [2001] methods were observed for the U11 (57%) and U14 (58%) years. Poor concordance between the Fels [1977] and Tanner-Whitehouse 3 [2001] methods is expected given the differences in reference samples used to develop each

method, the different bones and criteria used to assess skeletal maturity, and importantly the assignment of skeletal age given to a hand/wrist radiograph [27]. Given the relatively small sample size of the study ($n = 40$), we advise future studies use larger sample sizes whilst including players of different ethnicities to validate these findings, which is vital given that soccer academies across the world are becoming increasingly more diverse and consist of players with different ethnicities, who undergo different patterns of maturation [20,53]. A higher level of agreement was observed in the U11 and U14 years, which is interesting as typically both U11 and U14 years are considered pre-PHV and circa-PHV respectively [55]. Therefore, the higher levels of agreement in these age groups likely reflect the high proportion of average maturing players in these age groups. However, the lower levels of agreement in the other age groups are a possible reflection of the variance in the number of earlier and later maturing players in these age groups and therefore, individual timing and growth rates are important factors to be considered for players within these age groups [9].

The analysis of non-invasive methods to classify player maturity status revealed a higher amount of average maturing players using the Moore (2015; 50%) compared to the Mirwald (2002; 43%) method with a higher amount of later maturing players using the Mirwald (2002; 66%) compared to the Moore (2015; 43%) method. Typically, substantial agreement (64–67%) was observed between maturity offset methods, with only a moderate agreement (44–50%) seen between maturity offset and predicted adult height methods, utilising a more conservative threshold of 85–95% of predicted adult height. Interestingly, this level of agreement decreased when using a less conservative threshold of 88–93% of predicted adult height. The level of agreement was only moderate (58–60%) between maturity offset methods and fair (31–43%) between maturity offset and predicted adult height methods when using this less conservative threshold. The higher concordance between maturity offset methods is expected given they are all derived from an identical regression equation [7] with the predicted adult height equation deriving from an alternative regression equation. A lower agreement between maturity offset and predicted adult height methods may also be reflective of the different variables that these methods collect (adult height vs. time period from PHV). Together this re-iterates the premise that maturity offset and predicted adult height methods should not be used interchangeably [7]. The higher level of agreement when utilising a more conservative threshold is unsurprising, as they account for the error rate associated with assessments, as well as providing a broad range of players who are classified as on time in their maturity classification [17]. To sum, Salter *et al.*, [7] highlights the differences in the classification of players maturity status using various invasive and non-invasive methods. Practitioners are advised not to use these methods interchangeably and instead consider the individual maturational timing of players. Longitudinal studies with larger sample sizes are required to validate findings presented in this review.

## Limitations of included studies

The critical appraisal of the included studies revealed a higher proportion of moderate compared to high quality studies. A limitation of the current review is that we did not capture any randomised controlled trials and alternatively reviewed studies adopting observational designs. The studies included contained a higher proportion of cross-sectional ($n = 8$) compared to longitudinal (n = 5) studies which could be considered a limitation of the current review. Longitudinal studies have the potential to better describe the relationship between invasive and non-invasive methods over time which is not possible with cross-sectional studies. Longitudinal studies also have the capacity to determine the stability and reliability of some of the estimates and measures of maturation that are produced by these invasive and non-invasive methods [49]. A further assessment of the included studies is the inconsistency

in sample sizes, with a highly variable range of participants ($n$ = 17–1831). A limitation consistently reported from the authors of included studies was the underpowered sample sizes which may have potentially influenced some of the findings presented in this review. However, some of the included studies included predicted values for outcome variables such as age at PHV, maturity offset, predicted adult height, and biological age, however, observed values were absent for some of these variables and thus the true extent of the reliability of these estimates currently remains unknown. Results for some of the included studies are restricted to a limited number of age groups (e.g., number of skeletally mature players) hence some of the results may not be transferrable to other age groups. Collectively, the number of limitations associated with the included studies suggests the need for future research examining maturation status and timing in academy soccer players to consider study design and data capture procedures.

## Strengths and limitations of the review process

The present review included an objective framework (AMSTAR 2) to analyse the research process and subsequently reduce any bias within the review. The use of this framework revealed many strengths of the review process such as the inclusion criteria including all necessary components of PICOSS (Population, Intervention/Comparator, Outcome, Study design, Setting), with all relevant studies being captured using a comprehensive literature search, incorporating multiple databases. Further, we used a detailed screening process completed by two members of the research team and all included and excluded studies were justified by consensus. Data extraction was completed by two members of the research team which aided in the synthesis of individual studies based on common outcome variables, methods, and designs. An objective framework was used to assess study quality. The review process accounted for the risk of bias (e.g., publication bias) when interpreting the individual study findings.

The review process is not without its limitations, for example we only included studies in the English language and therefore it is possible that some relevant articles may not have been captured due to the filtering of English language search terms. Furthermore, we only included studies examining male, youth soccer players aged U9-U18 years, consequently excluding female soccer players, male amateur players and adolescents from the general population and players <U9 or >U18 years, which makes the findings from the review applicable to only a small proportion of male soccer players. The high heterogeneity in the data prevented a meta-analysis from being completed and the review is limited to a narrative synthesis of the data. A final consideration is the confirmation that this review is of moderate quality (see **Table 5**).

## Comparisons to other reviews

To our knowledge, no existing systematic review for assessing maturational status and timing in academy soccer players is available, making comparisons to the present review challenging. Nonetheless, two reviews have been conducted on the general population from adolescents and focused on methods to predict PHV (timing) but not methods to assess maturational status [36] and aimed to provide a critical narrative summary of the methods to assess maturational status and timing in adolescents [28].

The results of the present narrative review demonstrate that the Mirwald [2002] equation overestimated age at PHV by 0.6–0.9 years, a similar finding that was reported by Mills *et al.*, [36] who found in three studies that in the year before PHV, the Mirwald [2002] equation also overestimated age at PHV. However, an extended finding from Mills *et al.*, [36] was the increased accuracy of the Mirwald [2002] equation for predicting age at PHV when data was acquired three years prior to the actual age of PHV, which equated to age 11 years in boys. This finding was not reported in the present review and shows some promise for these

anthropometry-based methods [36]. Another common finding between the present review and Mills *et al.*, [36] was the high reliability scores of radiograph-based methods (i.e., MRI, X-Ray/DXA scanning) and anatomical surrogate measures. The present review differs from Mills *et al.*, [36] as the radiograph-based methods in this review investigated skeletal age and grade of fusion (maturity status) as opposed to age at PHV. However, the established reliability of radiographic methods can give practitioners the confidence to consider these methods.

The present review concluded that Tanner-Whitehouse 3 [2001] skeletal ages are consistently lower than corresponding Tanner-Whitehouse 2 [1983] derived skeletal ages among youth athletes aged 11–17 years. This review concludes that the difference can be as much as 0.97–1.07 years for young soccer players aged 11–17 years and would support the argument that Tanner-Whitehouse 2 [1983] should be used instead of the Tanner-Whitehouse 3 [2001] [28]. A noteworthy discrepancy between the present review and Malina *et al.*, [28] was maturity status classification. A reasonable concordance for maturity status classification of soccer players was reported for skeletal age and age at PHV, a comparison not made by any of the studies in the present review. This review reported only a moderate concordance between invasive skeletal age methods and maturity offset vs. predicted adult height methods. A substantial concordance was reported between maturity offset methods, which reduced to only a moderate agreement when a less conservative banding threshold was used (88–93% predicted adult height). This highlights a potential gap in the literature for future research to investigate regarding the agreement of maturity status classification between invasive and non-invasive methods, given the failure of the present review to address this.

## Applied implications

The implications of the present review can benefit practitioners when assessing maturational status and timing in academy soccer players. Although the findings reported in this review may not be generalisable to amateur male or female players, they highlight some important considerations for soccer clubs responsible for male academy players. Firstly, many of the non-invasive methods adopted by soccer academies were developed using populations that significantly differ in ethnicity, socioeconomic background and maturational status from modern academy players, which questions the reliability of using these methods in the target population. Saying that, in the absence of any viable alternatives, practitioners working in soccer academies are restricted to using these non-invasive methods or opt for more invasive methods involving medical scanning and subsequent radiation exposure for assessing maturational status and timing in academy players. Given the relatively poor concordance between invasive and non-invasive methods for assessing maturational status and timing in academy soccer players highlighted in this review and other reviews [8], it is recommended that practitioners avoid using these methods interchangeably. It is worth noting that all non-invasive methods have associated errors when applied to individual players, therefore, new predictive methods or modifications to existing equations are warranted that carefully consider the individual timing and rate of maturation amongst this culturally diverse and unique population [9].

## Suggestions for future studies

Despite a wealth of studies using the general adolescent population, the investigation of maturational assessments and associated performance effects within academy soccer is still in its infancy. From a holistic perspective, practitioners and researchers in this field may need to look beyond simply the methods they employ to assess maturational status and timing in academy soccer players and consider the wider implications of their choices on issues such as

injury risk. Recent work has highlighted an increase in injury risk and incidence around reported mean ages (i.e., 13–14 years) at PHV [11,14]. Furthermore, the growth spurt coupled with its maturity-associated variations are among some of the injury risk factors for the developing male athlete [10]. Collectively, these findings demonstrate the importance of using reliable methods to correctly assess a player's maturational status, given the subsequent impact this can have for training load management and injury risk around the time of PHV [8,11].

Given the high amount of heterogeneity in the available literature, future research should focus on the development of a homogeneous approach to data collection of maturity-related data and outcome variables during maturational research. Such data will enable a subsequent quantitative analysis to be completed, thus allowing researchers to better understand the reliability of these invasive and non-invasive methods. One possible solution to achieve a homogenous approach to future research within this area is to gain industry consensus on the rationale for professional clubs using specific types of maturational assessment methods when compared to alternatives. Gaining consensus on some of these areas could facilitate the collection of some common outcome variables, which could eventually facilitate the completion of a quantitative meta-analysis in this research area. Further training and education of academy staff who are responsible for the collection of maturity-related data from players across the academy may also be required to ensure reliable and accurate maturational data on an individual and group basis is recorded.

The findings presented here suggest the need for more longitudinal studies, given the excess of cross-sectional evidence, that are endorsed by governing bodies (e.g., the English Premier League and the English Football Association) and continue to utilise both invasive and non-invasive methods to monitor maturational status and timing amongst this large and ethnically diverse population. Given the amount of heterogeneity within the results, combined with largely moderate study quality, the true reliability of some of the most widely used methods to assess maturational status and timing in academy soccer players cannot be determined. It is imperative that the true reliability of these methods is established given the further implications of maturity on injury risk [10] and categorisation of academy soccer players for bio-banding [50].

## Conclusions

In this present review, we identified 15 studies that utilised invasive, non-invasive or a combination of both methods to assess maturational status and timing in academy soccer players. Despite the number of methods available to modern practitioners, no methods provided equivalent estimations of adult height, skeletal age, or age at PHV. Discrepancies were evident between actual and predicted adult height and actual vs predicted age at PHV. Practitioners utilising the Bayley-Pinneau [1952], Tanner-Whitehouse 2 [1983] or Khamis-Roche [1994] methods to predict adult height can be supported that these methods produce an estimated adult stature within 1cm of actual adult height. Similarly, for age at PHV, practitioners may utilise either the Moore [2015] equations or the Fransen [2018] equation in academy soccer players despite some recent criticism [54]. The Moore [2015] equations produced the closest estimates to actual age at PHV, however the Fransen [2018] equation correlated highly with actual age at PHV (>90%), even when the period between chronological age and age at PHV was large. Practitioners should also be aware of the significantly younger skeletal ages when using the Tanner-Whitehouse 3 [2001] assessment compared to 2 [1983] method and are therefore advised to use the latter method for assessing skeletal age. The poor concordance between invasive and non-invasive methods, despite high correlative values, is a recommendation to practitioners that these methods should not be used interchangeably for assessing

maturational status and timing in academy soccer players. However, to understand the reliability of these various types of methods, further research with improved study designs and reporting of consistent outcome variables are needed to create a homogenous approach to research in this field. Given the well documented association between injury risk and maturation, this review highlights the importance of using reliable and accurate methods to assess maturational status and timing within youth academy soccer players. This review demonstrates a bias towards single club studies [53]; therefore, it is our contention that better co-collaboration between clubs and performance staff such as sport scientists would help clubs develop and implement alternative strategies to counteract this ongoing problem.

## Acknowledgments

We would like to acknowledge the support provided by Jackie Fealey an institutional electronic database specialist for her expertise in the literature searching phase of this study.

## Author Contributions

**Investigation:** Joseph Sullivan.

**Methodology:** Joseph Sullivan.

**Resources:** Joseph Sullivan.

**Software:** Joseph Sullivan.

**Supervision:** Simon J. Roberts, John Mckeown, Martin Littlewood, Christopher McLaren-Towlson, Matthew Andrew, Kevin Enright.

**Validation:** Joseph Sullivan.

**Visualization:** Joseph Sullivan.

**Writing – original draft:** Joseph Sullivan.

**Writing – review & editing:** Joseph Sullivan, Christopher McLaren-Towlson, Matthew Andrew.

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
