## [Decision Letter · Decision Letter 0]

25 Apr 2023

PONE-D-23-06938Methods to predict the timing and status of biological maturation in male adolescent soccer players: A narrative systematic review.PLOS ONE

Dear Dr. Sullivan,

Thank you for submitting your manuscript to PLOS ONE. After careful consideration, we feel that it has merit but does not fully meet PLOS ONE’s publication criteria as it currently stands. Therefore, we invite you to submit a revised version (MINOR REVISION) of the manuscript that addresses the points raised during the review process.

We look forward to receiving your revised manuscript.

Kind regards,

Miguel Ángel Saavedra-García, Ph.D.

Academic Editor

PLOS ONE

Journal Requirements:

Reviewers' comments:

Reviewer's Responses to Questions

**Comments to the Author**

1. Is the manuscript technically sound, and do the data support the conclusions?

Reviewer #1: Yes

Reviewer #2: Partly

2. Has the statistical analysis been performed appropriately and rigorously? 

Reviewer #1: Yes

Reviewer #2: N/A

3. Have the authors made all data underlying the findings in their manuscript fully available?

Reviewer #1: Yes

Reviewer #2: Yes

4. Is the manuscript presented in an intelligible fashion and written in standard English?

Reviewer #1: Yes

Reviewer #2: Yes

5. Review Comments to the Author

Reviewer #1: Q1) Clear and logical methodology systematically set out according to the relevant statements (PRIMSA etc..)

Q2) Unable to pool data due to heterogeneity of studies, which is clearly stated. Other than that this study has statistical rigour.

Q3) The data has been provided as part of the manuscript and the also supporting information.

Q4) Very well written with excellent use of the English language.

Reviewer #2: The article is well focused, but perhaps the introduction is too long and at times moves away from the objective of the review.

In inclusion and exclusion criteria, perhaps it would be appropriate to indicate a reference to the PICOSS process. It would be appropriate to indicate that the search was focused on English.

The age of 18 years, as an end of age criterion, it would be appropriate to indicate why that age, since the maturation process in men may not have been completed.

Fig 1 should revise the bibliographic citation, and the tabulation of the right column of stage two. It would also be appropriate to revise the format of the tables. In table 3 it would be appropriate to include the n/a legend.

I cannot find the reference in the text of table 4, ....

in the conclusions and in the limitations a paragraph could be introduced concerning the influence of race/ethnicity for the equations.

after all the information described, I think that the section "applied implications" and "suggestions for future studies" should be more specific, as they are too general for the work done.

6. PLOS authors have the option to publish the peer review history of their article (what does this mean?). If published, this will include your full peer review and any attached files.

Reviewer #1: No

Reviewer #2: No

---

## [Author Response · Author response to Decision Letter 0]

5 May 2023

The authors would like to thank the reviewers and handling editor of the review for their constructive comments for the initial submission and for considering this re-submission for publication.

---

## [Decision Letter · Decision Letter 1]

23 May 2023

Methods to predict the timing and status of biological maturation in male adolescent soccer players: A narrative systematic review.

PONE-D-23-06938R1

Dear Dr. Sullivan,

We’re pleased to inform you that your manuscript has been judged scientifically suitable for publication and will be formally accepted for publication once it meets all outstanding technical requirements.

Kind regards,

Miguel Ángel Saavedra-García, Ph.D.

Academic Editor

PLOS ONE

Additional Editor Comments (optional):

Reviewers' comments:

Reviewer's Responses to Questions

**Comments to the Author**

1. If the authors have adequately addressed your comments raised in a previous round of review and you feel that this manuscript is now acceptable for publication, you may indicate that here to bypass the “Comments to the Author” section, enter your conflict of interest statement in the “Confidential to Editor” section, and submit your "Accept" recommendation.

Reviewer #1: All comments have been addressed

Reviewer #2: All comments have been addressed

2. Is the manuscript technically sound, and do the data support the conclusions?

Reviewer #1: Yes

Reviewer #2: Yes

3. Has the statistical analysis been performed appropriately and rigorously? 

Reviewer #1: Yes

Reviewer #2: Yes

4. Have the authors made all data underlying the findings in their manuscript fully available?

Reviewer #1: Yes

Reviewer #2: Yes

5. Is the manuscript presented in an intelligible fashion and written in standard English?

Reviewer #1: Yes

Reviewer #2: Yes

6. Review Comments to the Author

Reviewer #1: Great job the authors should be very proud looking forward to seeing the paper published.

Q1. All comments addressed - great work

Q2. Technically sound - great job all comments addressed

Q3. Statistical comments addressed

Q4.

Q5. Written to a high standard

Reviewer #2: The authors have correctly answered the comments made. The authors have adjusted the document to the different aspects of form. The manuscript is ready for publication.

7. PLOS authors have the option to publish the peer review history of their article (what does this mean?). If published, this will include your full peer review and any attached files.

Reviewer #1: **Yes: **Steven Jones

Reviewer #2: **Yes: **Helena

---

## [Editor Report · Acceptance letter]

30 May 2023

PONE-D-23-06938R1 

Methods to predict the timing and status of biological maturation in male adolescent soccer players: A narrative systematic review. 

Dear Dr. Sullivan:

I'm pleased to inform you that your manuscript has been deemed suitable for publication in PLOS ONE. Congratulations! Your manuscript is now with our production department. 

Kind regards, 

on behalf of

Dr. Miguel Ángel Saavedra-García 

Academic Editor

PLOS ONE